# Stillbirth in twin pregnancies in Stockholm County 2000–2021—A descriptive study

**Matilda Varli Hjärne[1], Cecilia Pegelow Halvorsen[1,2], Hanna Åmark[1,3]***

**1** Department of Clinical Science and Education, Södersjukhuset, Karolinska Institutet, Stockholm, Sweden, **2** Department of Neonatology, Sachs' Children and Youth hospital, Södersjukhuset, Stockholm, Sweden, **3** Department of Obstetrics and Gynecology, Södersjukhuset, Stockholm, Sweden

\* hanna.amark@regionstockholm.se

## Abstract

### Introduction

The incidence of stillbirth is higher in twin pregnancies compared with singleton pregnancies. Known risk factors for stillbirth in twin pregnancies include selective fetal growth restriction, twin-to-twin transfusion syndrome (TTTS) and preeclampsia. However, the full range risk factors and causes of stillbirth in twin pregnancies remain incompletely explored.

### Aim

The aim was to describe the incidence of stillbirth in twin pregnancies, identify associated risk factors, explore the causes of stillbirth, and examine pregnancy characteristics. The comparisons were made according to chorionicity and between twin and singleton stillbirths.

### Materials and methods

This descriptive register-based cohort study included 1978 singleton and 193 twin stillbirths in Stockholm County, from 2000 until 2021, using data from Stockholm Stillbirth Group and Swedish Pregnancy Register. Twin stillbirths with different chorionicity were compared with each other and with singleton stillbirths, respectively.

### Results

Twin pregnancies had a higher incidence of stillbirth compared with singleton pregnancies. Between 2000 and 2021, the incidence of singleton stillbirth decreased from 4.7 to 3.4 and the incidence of twin stillbirth decreased from 14.2 to 7.2 per 1000 births, respectively. Twin stillbirth occurred at an earlier gestational age, and fetuses in twin stillbirths had lower birth weight compared with singleton stillbirths. In monochorionic twin pregnancies, stillbirth was strongly associated with TTTS, identified as

**Data availability statement:** Data cannot be shared publicly for the safety of participants according to the Research Ethics Committee due to sensitive patient information. However, data may be shared after approval from the Regional Research Ethics Committee. An ethics application can be written following the instruction in this website https://etikprovning-smyndigheten.se/.

**Funding:** The author(s) received no specific funding for this work.

**Competing interests:** The authors have declared that no competing interests exist.

**Abbreviations:** BMI, body mass index; DCDA, dichorionic diamniotic; FGR, fetal growth restriction (previously IUGR, Intrauterine growth restriction); GW, gestational week; MCDA, monochorionic diamniotic; MCMA, monochorionic monoamniotic; sFGR, selective fetal growth restriction; SGA, small for gestational age; TAPS, twin anemia polycythemia sequence; TTTS, twin-to-twin transfusion syndrome.

the main cause of stillbirth. The causes of stillbirth among dichorionic pregnancies were similar with those of preterm singleton stillbirth, with placental insufficiency and fetal growth restriction, as the most common causes.

## Conclusions

The risk of stillbirth in twin pregnancies was more than twice as high as in singleton pregnancies, although both incidences decreased over time. Stillbirth among monochorionic twins was especially associated with the unique risk factor TTTS. Causes of stillbirth among dichorionic twin pregnancies were similar with those of preterm singleton stillbirth.

---

## Introduction

The current incidence of stillbirth in Sweden is approximately 3–4 per 1000 births [1]. The incidence has decreased during the last two decades, but at a slow pace [2]. Stillbirth is defined as a fetus born without signs of life from gestational week (GW) 22 + 0 [3,4]. In Stockholm County, there are approximately 400 multiple pregnancies each year out of a total of 29 000 livebirths [3]. Twin pregnancies are considered high-risk pregnancies with increased risk of preeclampsia, selective fetal growth restriction (sFGR), twin-to-twin transfusion syndrome (TTTS), infants born small for gestational age (SGA), miscarriage, and stillbirth [5–9]. Therefore, twin pregnancies are closely monitored according to regional guidelines [6,7,10].

Risk factors for antenatal stillbirth include high (>35 years) or low (<19 years) maternal age, smoking, high body mass index (BMI), fetal growth restriction (FGR), pregestational diabetes, and preeclampsia [2,11–14]. Twin pregnancies carry additional risk factors to consider, but the incidence of stillbirth has also been reported to depend on type of twin pregnancy. In monochorionic diamniotic twin pregnancies (MCDA), the incidence of stillbirth has been reported to be almost three times higher than in dichorionic diamniotic twins (DCDA) [8,15]. Additional risk factors associated with monochorionicity include TTTS, and twin anemia-polycythemia sequence (TAPS) [5,6,8,10,16]. TTTS is usually diagnosed before GW 24 + 0 with a prevalence of 10–15% in MC pregnancies and has been suggested to be the most common cause of stillbirth in MCDA pregnancies [6,8]. TTTS, FGR and sFGR are associated with a majority of stillbirths in MCDA, and almost half of stillbirths in DCDA [8]. Furthermore, previous research has shown that the chance of survival for at least one fetus with sFGR was 99.8% and for double survival 92.3%, whereas the chance of survival of both fetuses in absence of sFGR was 98.7% [17]. According to the same study, DCDA pregnancies with sFGR had a fivefold higher risk of stillbirth compared with those without sFGR [17].

For singleton pregnancies, common causes of stillbirth include placental insufficiency, chorioamnionitis, placental inflammation or infection, umbilical cord complications and fetal abnormalities [12,13,18]. The most common causes of stillbirth differ between term and preterm infants. Among singleton preterm stillbirths, placental

insufficiency and FGR are reported to be the most common causes, and in addition there is a higher proportion of pre-eclampsia and placental abruption [19]. In term stillbirth, the most common causes include infections, placental insufficiency and there is a higher proportion of umbilical cord complications [19]. There is a higher proportion of SGA among preterm stillbirth compared to term stillbirth [19]. A previous study found that the morbidity and mortality rates in preterm singleton pregnancies were similar to those of preterm twin pregnancies when corrected for gestational age [20].

Most studies on stillbirth in twin pregnancies have presented a limited number of cases, whereas in larger studies on stillbirth, twin pregnancies were often excluded [2,16,19,21,22]. The aim of this study was to describe risk factors and causes of stillbirth in twin pregnancies, based on chorionicity. Comparisons were also made with singleton stillbirths. The study was performed on data from Stockholm County.

## Materials and methods

This retrospective, register-based descriptive cohort study included stillbirths from Stockholm Stillbirth Database and Swedish Pregnancy Register from 2000 until 2021. The register data contain all stillbirths in Stockholm County from gestational week (GW) 22 + 0 in accordance with WHO's definition [23]. The Stockholm Stillbirth group was established in 1998 and includes obstetricians from all delivery wards in Stockholm County accompanied with perinatal pathologists. The group has reviewed all stillbirths in Stockholm County since 1998. In 2016, the Stockholm Stillbirth Database was linked to the Swedish Pregnancy Register, which is a certified national quality register [24]. To ensure a comprehensive case inclusion, the data was cross-checked with the National Board of Health and Welfare's register data, confirming the inclusion of virtually all cases of stillbirth. Information was collected from maternity care records, delivery ward health charts and medical journals as well as from histopathological examinations of the placenta and fetal autopsies performed at the department of pathology at Karolinska University Hospital, Huddinge, Stockholm. The Stockholm stillbirth classification was used to determine the primary and secondary causes of stillbirth as well as the degree of certainty of the primary cause of stillbirth, which was done in structured, consensus driven process [13].

Stillbirths between 1998 and 2000 were excluded due to incomplete data. The remaining stillbirths were categorized as singleton (n = 1978), twin (n = 193) or triplet (n = 5) pregnancies. Triplet pregnancies were excluded due to the small sample size (Fig 1). Twin pregnancies were subdivided into four groups based on chorionicity and amnionicity: DCDA (n = 96), MCDA (n = 86), MCMA (n = 4) and unknown chorionicity (n = 7) (Fig 1).

The twin stillbirths were compared with singleton stillbirths and MCDA stillbirths were compared with DCDA according to maternal and fetal characteristics and with cause and certainty of cause of fetal death. In addition, preterm singleton stillbirths were compared with DCDA stillbirths. This comparison was performed since term and preterm stillbirth were supposed to differ according to cause of stillbirth and most twin pregnancies are delivered early term [19,20].

### Variates

Maternal variables included maternal age at delivery, BMI based on self-reported height and measured weight from the first prenatal visit during the first trimester, smoking during the first trimester (yes/no), mother origin outside Sweden (yes/no), in vitro fertilization (yes/no), maternal complications such as gestational diabetes mellitus, gestational hypertension or preeclampsia, based on ICD codes O24.4, O13.9, and O14, respectively [8]. In some cases, preeclampsia was the main cause of stillbirth in accordance to Stockholm Stillbirth classification and Stockholm Stillbirth group, but as preeclampsia was also considered a risk factor, it was defined as both a risk factor and a potential cause of stillbirth [13]. SGA was defined as birthweight below the 10th percentile [25]. Fetal sex was registered upon birth. Gestational age at diagnosis of stillbirth was based on routine mid-trimester fetal ultrasound. From 2015, gestational age was based on an ultrasound performed in GW 11 + 0 to GW 13 + 6 if the fetal biparietal diameter was ≥ 21 mm. Ultrasound in GW 11–13 + 6 is not offered by routine in Sweden. If no ultrasound was performed, the first day of last menstrual period was used to calculate gestational age. The cause of stillbirth and the certainty of the cause of stillbirth was decided in a consensus driven process by

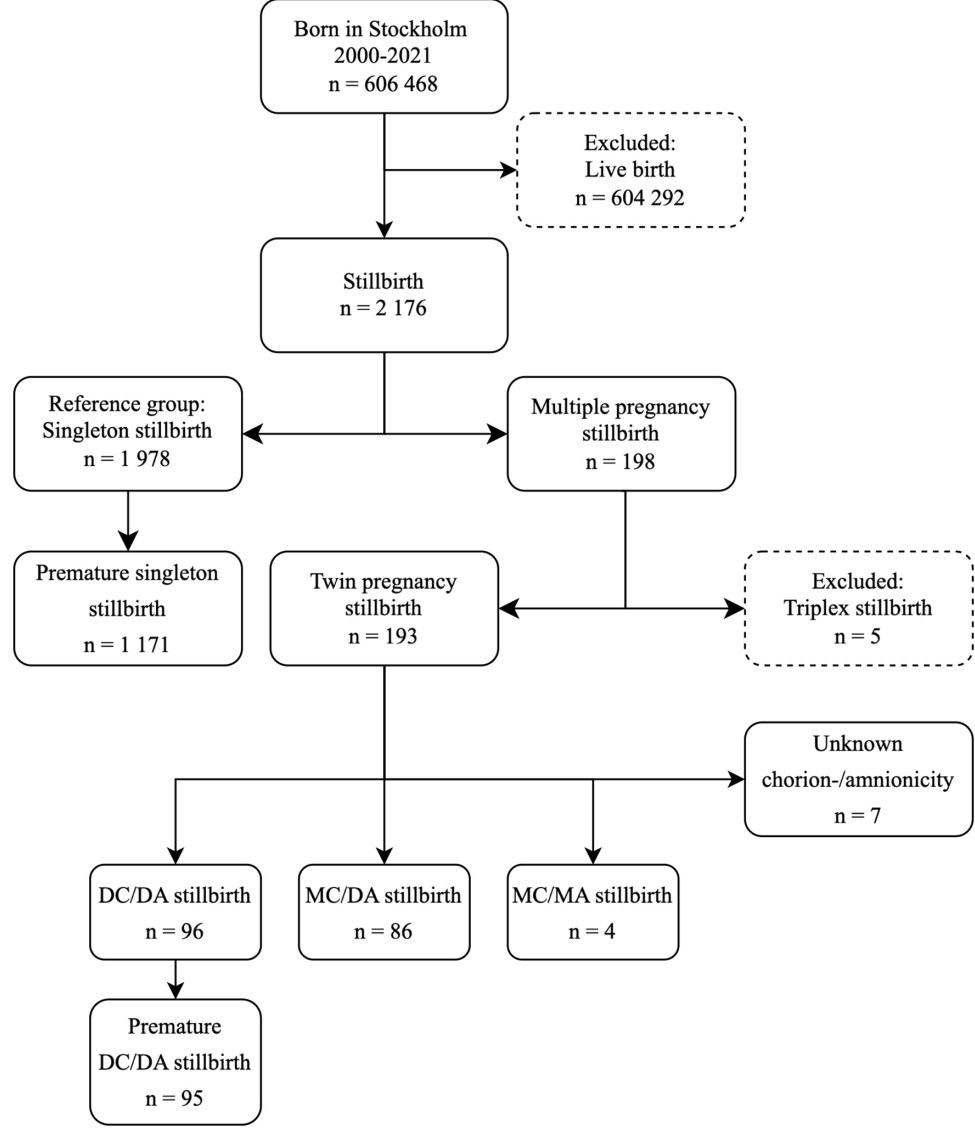

**Fig 1. Inclusion flowchart of stillbirths in Stockholm.** *Abbreviations*: GW, Gestational week; DCDA, Dichorionic diamniotic; MCDA, Monochorionic diamniotic; MCMA, Monochorionic monoamniotic. Stillbirth from GW 22 + 0 was included and divided into a reference group with singleton pregnancies. Multiple stillbirths were divided into triplet and twin stillbirths, the twins were then divided into DCDA, MCDA and MCMA, or unknown amnionicity and chorionicity.

Stockholm Stillbirth group according to the Stockholm Stillbirth classification [13]. Causative pathways according to fetal death was not studied in this setting.

## Statistics

The stillbirth incidence was calculated per year for singleton and for twin pregnancies per number of 1000 live births. Differences in incidences were evaluated using the chi-square test for multiple proportions. Categorical variables were compared with a chi-squared test and Student's t-test was used to analyze continuous variables. A p-value <0.05 was

considered statistically significant. Twin stillbirths were compared with singleton stillbirths, and DCDA stillbirths were compared with MCDA stillbirths regarding maternal and fetal characteristics, as well as cause and certainty of cause. Preterm singleton stillbirths were also compared with DCDA stillbirths. All data was analyzed using R Statistical Software version 4.2.1 (R Core Team 2021, Vienna, Austria).

Ethical approval for this study was obtained from the Swedish Ethical Review Authority Dnr 2020−01855 approved the 24th of June 2020 with amendment Dnr 2021−03412 approved the 9th of July 2021 concerning the possibility to cross-link the cases of stillbirth with Statistics Sweden and amendment Dnr 2022-03758-02 enabling a longer study period to include all cases of stillbirth up to 2030 approved the 3rd of August 2022. According to the ethical approval, anonymized information about women with stillbirths was obtained from the Stockholm Stillbirth database and the Swedish Pregnancy Register. These women were not asked for informed consent, since it was not required by the Research Ethics Committee. However, if a woman does not wish to participate in the Swedish Pregnancy Register, she may inform her midwife at the maternity clinic who will arrange for the data to be removed. All data was pseudonymized before analyses, where women and fetuses received an individual study number instead of using personal identity number. The researchers could not identify participating individuals. Data was accessed September 2022. The Research Ethics Committee prohibits data to be publicly available due to sensitive patient information. However, data will be shared after approval from the Regional Research Ethics Committee.

## Results

During the study period there were 1978 singleton and 193 twin reported stillbirths (Fig 1). The total incidence of singleton stillbirth in Stockholm from 2000 until 2021 decreased from 4.7 per 1000 in 2000–2004 to 3.3 per 1000 in 2018–2021. The incidence of stillbirth in twin pregnancies during the study period decreased from 12.7 per 1000 in 2000–2004 to 7.2 per 1000 live births in 2018–2021 (Fig 2).

A significantly shorter gestational age was observed in twin pregnancies compared with singleton pregnancies (194 days vs. 246 days; Table 1). There was a significantly higher proportion of in vitro fertilizations and of mothers originating from Sweden among twin stillbirths compared with singleton stillbirths. Maternal age was significantly higher among DCDA stillbirths compared with MCDA stillbirths and median gestational age at diagnosis of stillbirth was higher (201 vs. 177 days) in DCDA compared with MCDA stillbirths, (Table 2). DCDA stillbirths had a significantly higher proportion of in vitro fertilization compared with preterm singleton stillbirths (Table 3).

In twin pregnancies the overall most common cause of stillbirth was TTTS (29.6%), which is a unique event for MC twin pregnancies (Table 4). In twin stillbirth there was a significantly higher incidence of SGA compared with singleton stillbirth (56.7% vs. 38.0%; Table 4). Comparing DCDA and MCDA stillbirths there was a higher incidence of placental insufficiency in DCDA which was the most common cause of stillbirth among DCDA (37.5% vs. 11.6%; Table 5). The most common cause of stillbirth among MCDA was TTTS (55.8%; Table 5). MCDA stillbirths more often included both fetuses compared with DCDA stillbirths (59.7% vs 31.0%; Table 5). Comparing maternal characteristics and causes of stillbirth between DCDA and preterm singleton stillbirths showed a similar pattern (Table 6). A high incidence of SGA was found in both stillborn DCDA and stillborn preterm singleton cases (57.9 vs. 47.3%; Table 6).

Placental insufficiency was the most common cause of stillbirth among singleton pregnancies and was present more often in singleton stillbirths compared with twin stillbirths (30.1% vs 24.2%, Table 4). The second most common cause of stillbirth in singleton pregnancies was infection, which occurred to a larger proportion compared with twin stillbirths (19.5% vs 10.2%; Table 4).

## Discussion

In this retrospective register-based descriptive cohort study including stillbirths from Stockholm registers from 2000 until 2021, we showed that the risk of stillbirth was higher among twin pregnancies compared with singleton pregnancies.

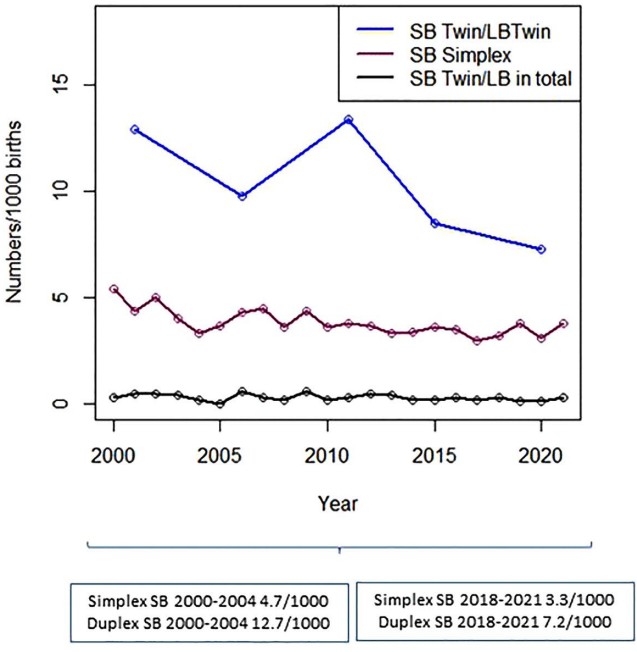

**Fig 2. Proportion of stillbirths in Stockholm among twin and singleton pregnancies.** *Abbreviations:* SB, Stillbirth; LB, Live birth. Stillbirths among twins/1000 twin births in Stockholm County (blue). Incidence of stillbirth in singleton pregnancies/1000 births (violet). Incidence of stillborn twins/1000 births (black). *P-values are significant at <0.05.

**Table 1. Maternal characteristics in stillborn singleton versus stillborn twins.**

|  | Stillborn Singleton n = 1978 | Stillborn Twins n = 193 | p-value* |
|---|---|---|---|
| Maternal age, years | 32.8 (29.8,36.8) | 33 (29.0,37.0) | 0.64 |
| BMI, kg/m2 | 24.2 (21.8,27.9) | 24.2 (21.5,27.4) | 0.36 |
| Nullipara (n, %) | 797 (44.8) | 72 (42.1) | 0.56 |
| Smoking (n, %) | 4 (1.8) | 13 (9.1) | <0.001 |
| Mother born in Sweden (n, %) | 884 (56.9) | 102 (67.1) | 0.02 |
| IVF (n, %) | 104 (5.3) | 35 (18.1) | <0.001 |
| Gestational age, days | 246 (194,273) | 194 (165,222) | <0.001 |
| GDM (n, %) | 34 (1.8) | 3 (1.6) | 0.87 |
| Gestational hypertension (n, %) | 75 (4.6) | 7 (4.4) | 1 |
| Preeclampsia (n, %) | 95 (5.1) | 11 (6.0) | 0.7 |

*Note*: Maternal characteristics were compared between stillbirths in singleton and twin pregnancies. Data is presented as median (range) or numbers (%).

*Abbreviations*: BMI, body mass index; IVF, in vitro fertilization; GDM gestational diabetes mellitus.

*p-values were considered significant p < 0.05.

**Table 2. Maternal characteristics in stillborn DCDA versus MCDA twins.**

|  | DCDA n=96 | MCDA n=86 | p-value* |
|---|---|---|---|
| **Maternal age, years** | 34.0 (30.8,37.0) | 31.2 (27.1,36.0) | 0.01 |
| **BMI, kg/m2** | 24.6 (21.1,28.6) | 23.8 (21.7,26.8) | 0.78 |
| **Nullipara (n, %)** | 41 (49.4) | 30 (38.5) | 0.22 |
| **Smoking, (n, %)** | 6 (6.2) | 7 (8.1) | 0.86 |
| **Mother born in Sweden (n, %)** | 47 (60.3) | 51 (73.9) | 0.12 |
| **IVF (n, %)** | 20 (20.8) | 13 (15.1) | 0.42 |
| **Gestational age, days** | 201 (175,251) | 177 (159,205) | <0.001 |
| **GDM (n, %)** | 3 (3.2) | 0 (0) | 0.29 |
| **Gestational hypertension (n, %)** | 6 (8.1) | 2 (2.6) | 0.24 |
| **Preeclampsia (n, %)** | 9 (9.6) | 3 (3.6) | 0.20 |

*Note*: Maternal characteristics were compared between stillbirths in dichorionic diamniotic (DCDA) and monochorionic diamniotic (MCDA) twin pregnancies. Data is presented as median (range) or numbers (%).

*Abbreviations*: BMI, body mass index; IVF, in vitro fertilization; GDM gestational diabetes mellitus; DCDA, Dichorionic diamniotic; MCDA, monochorionic diamniotic.

*p-values were considered significant p<0.05.

**Table 3. Maternal characteristics in stillborn preterm singleton versus stillborn preterm DCDA.**

|  | Preterm singleton n=1171 | Preterm DCDA n=95 | p-value* |
|---|---|---|---|
| **Maternal age, years** | 32.9 (29.8,36.8) | 34 (31.0,37.0) | 0.19 |
| **BMI, kg/m2** | 24.1 (21.6,27.6) | 24.6 (21.1,28.6) | 0.61 |
| **Nullipara (n, %)** | 469 (44.3) | 40 (48.78) | 0.51 |
| **Smoking, (n, %)** | 2 (0.2) | 6 (5.9) | <0.001 |
| **Mother born in Sweden (n, %)** | 506 (54.9) | 47 (60.3) | 0.43 |
| **IVF (n, %)** | 66 (5.7) | 20 (21.1) | <0.001 |
| **Gestational age, days** | 204 (175,236) | 199 (174,251) | 0.46 |
| **GDM (n, %)** | 16 (1.4) | 3 (3.2) | 0.36 |
| **Gestational Hypertension (n, %)** | 53 (5.5) | 5 (6.9) | 0.82 |
| **Preeclampsia (n, %)** | 81 (7.2) | 8 (8.6) | 0.78 |

*Note*: Maternal characteristics were compared between preterm stillbirths in singleton and dichorionic diamniotic (DCDA) twin pregnancies. Data is presented as median (range) or numbers (%).

*Abbreviations*: BMI, body mass index; IVF, in vitro fertilization; GDM gestational diabetes mellitus; DCDA, Dichorionic diamniotic.

*p-values were considered significant p<0.05.

Stillbirths occurred earlier in twin vs. singleton pregnancies and earlier in MCDA vs. DCDA pregnancies. Twin stillbirth fetuses were more often SGA compared with singleton fetuses. The increased risk of stillbirth in MCDA pregnancies was predominantly related to the incidence of TTTS. Regarding maternal characteristics and causes of stillbirth, DCDA stillbirths were found comparable to singleton preterm stillbirths. Hence, comparing singleton stillbirth with twin stillbirths may highlight differences which could be caused by gestational length.

The proportion of MCDA twin stillbirths may have been affected by a national assignment of TTTS fetoscopic ablation treatment to Karolinska University hospital in Stockholm County from 2001. TTTS was found to be the most common cause of stillbirth in twin pregnancies, while placental insufficiency was the most common cause of stillbirth among singleton pregnancies, consistent with findings in previous studies [22]. The most common cause of stillbirth among DCDA was placental insufficiency, which was more common than in MCDA stillbirths. However, there is no reason to conclude that

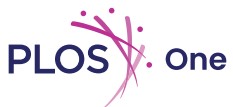

**Table 4. Fetal characteristics and causes of stillbirth in stillborn singleton versus stillborn twins.**

|  | Singleton n = 1978 | Twins n = 193 | p-value* |
|---|---|---|---|
| Fetal weight (grams) | 2247.5 (829,3129) | 690 (435,1580) | <0.001 |
| SGA 10th percentile (n, %) | 733 (38.0) | 94 (56.7) | <0.001 |
| Fetal sex, female (n, %) | 981 (49.6) | 88 (45.6) | 0.32 |
| TTTS (n, %) | 0 (0) | 51 (27.4) | n/a |
| Immunization (n, %) | 3 (0.2) | 0 (0) | 1 |
| Asphyxia (n, %) | 12 (0.6) | 0 (0) | 0.62 |
| Malformation (n, %) | 167 (8.9) | 11 (5.9) | 0.22 |
| Infection (n, %) | 369 (19.5) | 19 (10.2) | 0.003 |
| Placental insufficiency/FGR (n, %) | 568 (30.1) | 45 (24.2) | 0.11 |
| Umbilical cord complications (n, %) | 116 (6.1) | 7 (3.8) | 0.25 |
| Placental abruption (n, %) | 167 (8.9) | 3 (1.6) | <0.001 |
| Preeclampsia (n, %) | 100 (5.3) | 12 (6.5) | 0.62 |
| Diabetes (n, %) | 36 (1.9) | 0 (0) | 0.07 |
| Cholestasis (n, %) | 14 (0.7) | 0 (0) | 0.63 |
| Other cause of stillbirth (n, %) | 41 (2.1) | 16 (8.6) | <0.001 |
| Cause of stillbirth un-known (n, %) | 233 (12.3) | 20 (10.8) | 0.61 |

*Note*: Fetal characteristics and causes of stillbirth were compared between stillborn singletons and twins. Data is presented as median (range) or numbers (%).

*Abbreviation*: SGA, small for gestational age; TTTS, twin to twin transfusion syndrome; FGR, fetal growth restriction.

*p-values were considered significant p < 0.05.

**Table 5. Fetal characteristics and causes of stillbirth in stillborn DCDA versus stillborn MCDA.**

|  | DCDA n = 96 | MCDA n = 86 | p-value* |
|---|---|---|---|
| Fetal weight (grams) | 730 (400,1988) | 635 (466,1138) | 0.68 |
| SGA 10th percentile (n, %) | 44 (57.1) | 43 (55.1) | 0.93 |
| Fetal sex, female (n, %) | 42 (43.8) | 40 (46.5) | 0.82 |
| Cases with two stillborn fetuses (n, %) | 26 (31.0) | 46 (59.7) | <0.001 |
| TTTS (n, %) | 0 (0) | 48 (55.8) | n/a |
| Malformation (n, %) | 6 (6.8) | 4 (4.7) | 0.77 |
| Infection (n, %) | 17 (19.3) | 2 (2.3) | <0.001 |
| Placental insufficiency/FGR (n, %) | 33 (37.5) | 10 (11.6) | <0.001 |
| Umbilical cord complications (n, %) | 2 (2.0) | 2 (2.3) | 1 |
| Preeclampsia (n, %) | 8 (9.1) | 4 (4.7) | 0.39 |
| Other cause of stillbirth (n, %) | 11 (12.5) | 2 (2.3) | 0.02 |
| Cause of stillbirth un-known (n, %) | 9 (10.2) | 11 (12.8) | 0.77 |

*Note*: Fetal characteristics and causes of stillbirth were compared between dichorionic diamniotic (DCDA) and monochorionic diamniotic (MCDA) twins. Data is presented as median (range) or numbers (%).

*Abbreviation*: SGA, small for gestational age; TTTS, twin to twin transfusion syndrome; FGR, fetal growth restriction; DCDA, Dichorionic diamniotic; MCDA, monochorionic diamniotic.

*p-values were considered significant p < 0.05.

MCDA, compared with DCDA stillbirths, have a lower degree of placental insufficiency. The reported difference is probably due to the high number of stillbirths caused by TTTS among MCDA. A study by Mahony et al. [8], showed that stillbirth in MC pregnancies due to TTTS most often occurs before GW 30+0, a trend also observed in this study [8].

**Table 6. Fetal characteristics and causes of stillbirth in stillborn singleton preterm versus stillborn preterm DCDA.**

| | Singleton preterm n = 1171 | Preterm DCDA n = 95 | p-value* |
|---|---|---|---|
| **Fetal weight (grams)** | 1055 (545,2000) | 727.5 (400,1950) | 0.01 |
| **SGA 10th percentile (n, %)** | 541 (47.3) | 44 (57.9) | 0.09 |
| **Fetal sex, female (n, %)** | 585 (50.0) | 41 (43.2) | 0.13 |
| **Immunization (n, %)** | 3 (0.3) | 0 (0) | 1 |
| **Asphyxia (n, %)** | 5 (0.5) | 0 (0) | 1 |
| **Malformation (n, %)** | 123 (11.0) | 6 (6.8) | 0.30 |
| **Infection (n, %)** | 174 (15.6) | 17 (19.3) | 0.44 |
| **Feto-maternal transfusion (n, %)** | 21 (1.9) | 0 (0) | 0.40 |
| **Placental insufficiency/IUGR (n, %)** | 337 (30.2) | 33 (37.5) | 0.19 |
| **Umbilical cord complications (n, %)** | 47 (4.2) | 2 (2.3) | 0.55 |
| **Placental abruption (n, %)** | 110 (9.9) | 0 (0) | 0.0006 |
| **Preeclampsia (n, %)** | 88 (7.9) | 8 (9.1) | 0.84 |
| **Diabetes (n, %)** | 17 (1.5) | 0 (0) | 0.49 |
| **Cholestasis (n, %)** | 9 (0.8) | 0 (0) | 0.84 |
| **Other cause of stillbirth (n, %)** | 32 (2.9) | 11 (12.5) | <0.001 |
| **Cause of stillbirth un-known (n, %)** | 136 (12.2) | 9 (10.2) | 0.71 |

*Note*: Fetal characteristics and causes of stillbirth were compared between preterm stillborn singletons and preterm stillborn dichorionic diamniotic (DCDA) twins. Data is presented as median (range) or numbers (%).

*Abbreviation*: SGA, small for gestational age; TTTS, twin to twin transfusion syndrome; FGR, fetal growth restriction; DCDA, Dichorionic diamniotic.

*p-values were considered significant p < 0.05.

Cause of stillbirth due to FGR and placental insufficiency varies according to the stillbirth rate and income level of the country. While it is an important factor in high-middle income countries it is not among the top five causes in low-income countries [26]. FGR is an important risk factor for stillbirth in both twin and singleton gestations [27]. Thus, some kind of screening for FGR is recommended [28,29] and FGR/placental insufficiency will appear among the more common causes when the most preventable risks are addressed [28].

DCDA fetuses, as well as singleton fetuses, have their own placentas. The DCDA placentas do not share vessels and cannot develop TTTS. However, the size of each placenta may differ which may cause a difference in fetal size. Placental insufficiency was more prevalent in DCDA stillbirths than in the whole group of singleton stillbirths, supported by previous studies [8,22]. For twin pregnancies in Stockholm, it is recommended to induce labor at GW 38 + 0 for DCDA, and at GW 37 + 0 for MCDA pregnancies, respectively [6,7]. Comparing the incidence of placental insufficiency in preterm DCDA and preterm singleton stillbirths showed no significant difference. Placental insufficiency was the most common cause of stillbirth among both DCDA and preterm singleton stillbirths. Infections were the second most common cause of stillbirth in preterm singleton and DCDA pregnancies.

There was a higher proportion of SGA in twin compared with singleton stillbirth. More than half of the DCDA and MCDA stillbirths were SGA, compared with singleton stillbirths. In preterm singleton stillbirths, the incidence of SGA was more similar to the twin pregnancies. Previous studies have shown that live born infants from twin pregnancies are generally smaller than live born infants from singleton pregnancies [30]. Hence, the larger proportion of SGA among twin stillbirths compared with singleton stillbirths may reflect the difference comparing twins and singletons regardless of livebirth or stillbirth.

Previous research suggested that if cases of TTTS and sFGR were excluded, the causes of stillbirth would be largely similar between DCDA and MCDA pregnancies [8]. Another study showed an association between high birthweight discordance in twins and FGR, but showed no correlation with mortality [31]. According to our results, the risk of stillbirth

among DCDA is increased because the placentas do not always have the capacity for supporting both fetuses and due to this there is a larger risk of FGR or sFGR. Among MCDA pregnancies, the increased risk of stillbirth is most often due to TTTS. If the risk of TTTS and sFGR/FGR could be eliminated, the risk of stillbirth would probably be comparable to that of singleton pregnancies. In a future study, an adjustment for cofounding factors such as TTTS and sFGR could be made to determine if this would yield a different result.

A strength of this study was the large number of stillbirths; 193 twin stillbirths were included. Since twin pregnancies are uncommon and stillbirth is even rarer, this study comprising a substantial number of stillbirths, provided a greater possibility to draw conclusions compared with previous studies. Furthermore, available data on type of twin pregnancy was essential to draw conclusions according to chorionicity.

Noteworthy is that, according to National Board of Social Affairs and Health, maternal characteristics have changed during the study period [3]. The proportion of pregnant smokers has decreased while BMI and maternal age have increased [3]. The criteria of the maternal diagnoses preeclampsia and gestational diabetes have changed during the study period. The criteria have become wider, and a larger proportion of pregnant women received these diagnoses in 2021 compared with 2019 for preeclampsia, and with 2018 for gestational diabetes [32,33].

A potential source of selection bias could be that Karolinska University hospital in Stockholm received the national assignment of fetoscopic laser surgery for treatment of TTTS in 2001. Thereafter, all Swedish pregnant women who required interventions for TTTS were and are referred to Karolinska University hospital in Stockholm County. Due to increased risks associated with TTTS, but also risks associated with the treatment, this could have had an impact on the number of stillbirths in MC twin pregnancies of the studied cohort of Stockholm County.

## Conclusions

Although stillbirth incidence has decreased, twin pregnancies remain at higher risk than singletons due to unique factors. In MC pregnancies, TTTS is the primary risk factor, while sFGR significantly increases stillbirth risk across all twin types, indicating FGR as a key mediator. Furthermore, causes of preterm DCDA stillbirth are consistent with preterm singleton stillbirth.

## Acknowledgments

We would like to thank Stockholm Stillbirth group for sharing essential data used in this study.

## Author contributions

**Conceptualization:** Matilda Varli Hjärne, Cecilia Pegelow Halvorsen, Hanna Åmark.

**Data curation:** Matilda Varli Hjärne, Hanna Åmark.

**Methodology:** Hanna Åmark.

**Supervision:** Cecilia Pegelow Halvorsen, Hanna Åmark.

**Writing – original draft:** Matilda Varli Hjärne.

**Writing – review & editing:** Cecilia Pegelow Halvorsen, Hanna Åmark.

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
