## [Decision Letter · Decision Letter 0]

25 Aug 2025

Dear Dr. Åmark,

We look forward to receiving your revised manuscript.

Kind regards,

Sikolia Wanyonyi

Academic Editor

PLOS ONE

Journal Requirements:

For additional information about PLOS ONE ethical requirements for human subjects research, please refer to http://journals.plos.org/plosone/s/submission-guidelines#loc-human-subjects-research .

4. Please include captions for your Supporting Information files at the end of your manuscript, and update any in-text citations to match accordingly. Please see our Supporting Information guidelines for more information: http://journals.plos.org/plosone/s/supporting-information .

Additional Editor Comments:

The reviewers have recommended minor revisions. Please address them and resubmit the manuscript for consideration.

Reviewers' comments:

Reviewer's Responses to Questions

**Comments to the Author**

1. Is the manuscript technically sound, and do the data support the conclusions?

Reviewer #1: Yes

Reviewer #2: Yes

Reviewer #3: Yes

2. Has the statistical analysis been performed appropriately and rigorously?

Reviewer #1: Yes

Reviewer #2: Yes

Reviewer #3: Yes

3. Have the authors made all data underlying the findings in their manuscript fully available?

Reviewer #1: Yes

Reviewer #2: No

Reviewer #3: No

4. Is the manuscript presented in an intelligible fashion and written in standard English?

Reviewer #1: Yes

Reviewer #2: Yes

Reviewer #3: Yes

Reviewer #1: This is a retrospective descriptive study of still births in twin pregnancies between 2000 and 2021, using data from Stockholm stillbirth group 10 and Swedish Pregnancy Register. The authors’ aim was to examine pregnancy characteristics, and determine the incidence, risk factors and causes of still birth in twin pregnancies.

The abstract and introduction to the research topic were well written and relevant. The methodology was well described, and appropriate statistical analysis was made. Comparisons were made between the singleton and twins still births according to the chorionicity of the twins. The results were satisfactorily presented with figures and tables. The discussion and conclusions were relevant to the results provided. The strength and limitations to the study were provided.

This study further educates the readers on the influence of chorionicity of twin pregnancy on stillbirths and advances existing literature on the topic.

Reviewer #2: The manuscript is well written and documented, the undocumented aspect was clearly stated by the author, and it was due to incomplete data. I commend the work, it is an excellent contribution to knowledge and it will serve as a good reference for future research in this category.

Reviewer #3: This is an interesting paper highlighting stillbirth in the area. However authors should make their discussions more robust by comparing findings from their study with previous study. For example. the finding that placental insufficiency was strongly associated with stillbirths in singleton and dichorionic twins should be compared with other findings in the subregion and other climes.

**Do you want your identity to be public for this peer review?** For information about this choice, including consent withdrawal, please see our Privacy Policy

Reviewer #1: No

Reviewer #2: **Yes: ** Akinyosoye Ajiboye

Reviewer #3: **Yes: ** Dr. Emmanuel Ajuluchukwu Ugwa PhD FWACS FMCOG

---

## [Author Response · Author response to Decision Letter 1]

18 Oct 2025

Dear Editor och Reviewers,

Thank you for considering our manuscript and providing us with constructive comments to improve this paper. We have answered all comments in a point-to-point fashion below.

Reviewer #1: This is a retrospective descriptive study of still births in twin pregnancies between 2000 and 2021, using data from Stockholm stillbirth group 10 and Swedish Pregnancy Register. The authors’ aim was to examine pregnancy characteristics, and determine the incidence, risk factors and causes of still birth in twin pregnancies.

The abstract and introduction to the research topic were well written and relevant. The methodology was well described, and appropriate statistical analysis was made. Comparisons were made between the singleton and twins still births according to the chorionicity of the twins. The results were satisfactorily presented with figures and tables. The discussion and conclusions were relevant to the results provided. The strength and limitations to the study were provided.

This study further educates the readers on the influence of chorionicity of twin pregnancy on stillbirths and advances existing literature on the topic.

We thank the reviewer for your kind words.

Reviewer #2: The manuscript is well written and documented, the undocumented aspect was clearly stated by the author, and it was due to incomplete data. I commend the work, it is an excellent contribution to knowledge and it will serve as a good reference for future research in this category.

We thank the reviewer for your kind words.

Reviewer #3: This is an interesting paper highlighting stillbirth in the area. However, authors should make their discussions more robust by comparing findings from their study with previous study. For example. the finding that placental insufficiency was strongly associated with stillbirths in singleton and dichorionic twins should be compared with other findings in the subregion and other climes.

Thanks, this is an important comment and we have added to the discussion: “Cause of stillbirth due to FGR and placental insufficiency varies according to the stillbirth rate and income level of the country. While it is an important factor in high-middle income countries it is not among the top five causes in low-income countries[26].FGR is an important risk factor for stillbirth in both twin and singleton gestations [27]. Thus, some kind of screening for FGR is recommended [28, 29] and FGR/placental insufficiency will appear among the more common causes when the most preventable risks are addressed[28].”

---

## [Editor Report · Decision Letter 1]

12 Nov 2025

Stillbirth in twin pregnancies in Stockholm County 2000-2021 - a descriptive study

PONE-D-25-23412R1

Dear Dr. Åmark,

We’re pleased to inform you that your manuscript has been judged scientifically suitable for publication and will be formally accepted for publication once it meets all outstanding technical requirements.

Kind regards,

Sikolia Wanyonyi

Academic Editor

PLOS ONE

Additional Editor Comments (optional):

Thank you for addressing the comments raised by the reviewers.
---

## [Editor Report · Acceptance letter]

PONE-D-25-23412R1

PLOS ONE

Dear Dr. Åmark,

I'm pleased to inform you that your manuscript has been deemed suitable for publication in PLOS ONE. Congratulations! Your manuscript is now being handed over to our production team.

Kind regards,

on behalf of

Dr. Sikolia Wanyonyi

Academic Editor

PLOS ONE